# An Anti-Nucleocapsid Antigen Sars-Cov-2 Total Antibody Assay Finds Comparable Results in Edta-Anticoagulated Whole Blood Obtained from Capillary and Venous Blood Sampling

**Martin Risch [1,*], Marc Kovac [2], Corina Risch [2], Dorothea Hillmann [2], Michael Ritzler [2], Nadia Wohlwend [2], Thomas Lung [2], Michael Allmann [2], Christoph Seger [2] and Lorenz Risch [2,3]** 

[1]   Zentrallabor, Kantonsspital Graubünden, 7000 Chur, Switzerland
[2]   Labormedizinisches Zentrum Dr. Risch, Wuhrstrasse 14, 9490 Vaduz, Liechtenstein;
      marc.kovac@risch.ch (M.K.); corina.risch@risch.ch (C.R.); dorothea.hillmann@risch.ch (D.H.);
      michael.ritzler@risch.ch (M.R.); nadia.wohlwend@risch.ch (N.W.); thomas.lung@risch.ch (T.L.);
      michael.allmann@risch.ch (M.A.); christoph.seger@risch.ch (C.S.); lorenz.risch@risch.ch (L.R.)
[3]   Faculty of Medical Sciences, Private Universität im Fürstentum Liechtenstein, Dorfstrasse 24,
      9495 Triesen, Liechtenstein
[*]   Correspondence: martin.risch@ksgr.ch; Tel.: +41-81-2566530

**Abstract:** Although SARS-CoV-2 antibody assays have been found to provide valid results in EDTA-anticoagulated whole blood, so far, they have not demonstrated that antibody levels in whole blood originating from capillary blood samples are comparable to antibody levels measured in blood from a venous origin. Here, blood is drawn simultaneously by capillary and venous blood sampling. Antibody titers are determined by an assay employing electrochemiluminescence (ECLIA) and SARS-CoV-2 total immunoglobulins are detected with specificity directed against the nucleocapsid antigen. Six individuals with confirmed COVID-19 and six individuals without COVID-19 are analyzed. Antibody titers in capillary venous whole blood did not show significant differences, and when corrected for hematocrit, they did not differ from the results obtained from serum. In conclusion, capillary sampled EDTA-anticoagulated whole blood seems to be an attractive alternative matrix for the evaluation of SARS-CoV-2 antibodies when employing ECLIA for detecting total antibodies directed against nucleocapsid antibodies.

**Dataset:** The dataset is submitted as a supplement to this paper in the journal Data.

**Dataset License:** CC BY 4.0.

**Keywords:** antibody; COVID-19; preanalytics; SARS-CoV-2; serology; whole blood

---

## 1. Summary

Laboratory analysis to diagnose acute or past COVID-19 has become central in mastering the current COVID-19 pandemic [1]. Whereas molecular diagnosis determining viral RNA of the SARS-CoV-2 virus represents the cornerstone to diagnose acute disease in the medical care of individual patients, the serological investigation of antibodies is carried out to provide several other indications which concentrate on public health [1,2]. SARS-CoV-2 antibody testing is mainly employed as a public health measure to monitor disease prevalence and to identify individuals potentially serving as convalescence plasma donors after COVID-19 has been cured [3,4]. In clinical medicine, serology is

also used for individuals showing an implausible negative result in molecular SARS-CoV-2 virus testing despite a high clinical suspicion of acute disease, and in individuals with COVID-19 symptoms in their history for which the acute phase, for various reasons, could not be investigated by diagnostic testing (e.g., lacking access to testing, mild symptoms not leading to medical consultation, etc.) [5,6].

As the pandemic is affecting many individuals worldwide, the need for tests can be considered to be large, both from the perspectives of clinical medicine and public health. Since most automated laboratory testing formats for SARS-CoV-2 antibodies are designed to measure the antibodies from plasma or serum, venous blood is usually the preferred method for drawing blood. Due to its complexity, venous blood drawing is mainly carried out in medical settings (e.g., physicians' offices or hospitals) [7]. During the COVID-19 pandemic, venous blood drawing can be considered problematic, because this may lead to the unnecessary occupancy of medical institutions by healthy individuals, despite the fact that these institutions have originally been designated to provide healthcare to sick persons. This burden arises from two potentially large collectives of individuals having recovered from COVID-19, i.e., individuals tested for determining the prevalence of COVID-19 and individuals with prior asymptomatic or undiagnosed COVID-19. Since capillary blood sampling is less complex than venous blood sampling, it is often carried out outside of a healthcare context and even has been demonstrated to be a safe procedure when individuals perform capillary blood sampling on themselves [8,9]. Therefore, capillary blood sampling would represent an attractive alternative for collecting a large volume of blood designated for SARS-CoV-2 antibody testing [10].

Capillary blood sampled as anticoagulated whole blood offers the possibility of providing larger sample volumes to automated laboratory analyzers when compared to serum or plasma. Even if it has been shown that SARS-CoV-2 antibodies in some assays can be reliably measured in whole blood, so far, it is not known whether antibody titers obtained with methods different from immunochromatographic lateral flow tests in capillary whole blood would be different from the respective antibody titers in venous whole blood [10,11]. Accordingly, the Infectious Diseases Society of America (IDSA) Guidelines on the Diagnosis of COVID-19 recently acknowledged a knowledge gap in terms of whether capillary or venous blood sampling would be preferable for COVID-19 serological testing [12]. A recent paper demonstrated that measurements from EDTA-anticoagulated whole blood provided SARS-CoV-2 N-antigen antibody test results which were comparable to the results obtained from serum testing [10]. However, the results of this investigation were obtained from venous blood and it remains an open question whether capillary and venous whole blood samples provide comparable results. Another study found that results for an ELISA assay detecting IgG antibodies directed against the receptor binding domain of the S1 protein are comparable between serum from venous blood drawing and dried blood spots obtained from capillary blood sampling [9]. Furthermore, data from other studies measuring antibody titers for other viral diseases such as measles or rubella suggest that results from capillary and venous blood sampling are comparable [13,14].

Therefore, we assembled data from a small collective that simultaneously underwent capillary and cubital venous blood sampling. Samples were collected as capillary EDTA-anticoagulated whole blood, as well as venous EDTA-anticoagulated whole blood and serum. The samples were analyzed with an electrochemiluminescence (ECLIA) method to detect SARS-CoV-2 nucleocapsid total antibodies. The data were gathered in September 2020 within the framework of the COVI-GAPP study, investigating the aptitude of a sensory bracelet to better recognize COVID-19 [15]. Data were stored with Microsoft Excel (Microsoft, Seattle, USA) and statistical analysis was carried out by means of MedCalc version 18.11.3 (MedCalc Software Ltd., Ostend, Belgium).

## 2. Data Description

The data contain demographic information and laboratory results from the participants, as well as results from SARS-CoV-2 total antibody testing by ECLIA for several matrices. A patient identifier (ID) was given in an anonymized form as a continuous number, sex was given as F (female) or M (male), and age was given in years. Participants with prior COVID-19 were designated with

1, whereas participants without COVID-19 were designated with 0. Hematocrit (HCT) was given as a percentage. Results from antibody testing were given as the cut-off index (COI) for venous EDTA-anticoagulated whole blood (VWB), capillary EDTA-anticoagulated whole blood (CWB), and venous serum (VS). The results from the EDTA-anticoagulated whole blood of capillary (CWBC) and venous blood (VWBC) were corrected for hematocrit.

Table 1 summarizes the participant characteristics and illustrates that none of the participants needed hospitalization. The participants with COVID-19 had a mild course and all experience a full clinical recovery.

**Table 1.** Participant characteristics. Continuous variables are given as median and interquartile range (IQR); n = number.

| Characteristic | COVID-19 | No COVID-19 |
|---|---|---|
| Particpiants (n) | 6 | 6 |
| Age (years) | 44 (39,46) | 43 (19,52) |
| Female/Male (n) | 4/2 | 4/2 |
| Hematocrit (%) | 44 (38,47) | 43 (41,45) |
| Hospitalized (n) | 0 | 0 |

There was no statistically significant difference between the SARS-CoV-2 results obtained in capillary and venous whole blood ($p = 0.44$). Analogously, there was no statistical difference between the whole blood results (i.e., capillary whole blood, $p = 0.73$; venous whole blood, $p = 0.42$) when corrected for hematocrit and the results obtained for serum. Figure 1 displays the results (without correction for hematocrit) obtained for capillary and venous whole blood. As the number of samples is limited, minor differences between capillary and venous blood might go unnoticed due to statistical type II error.

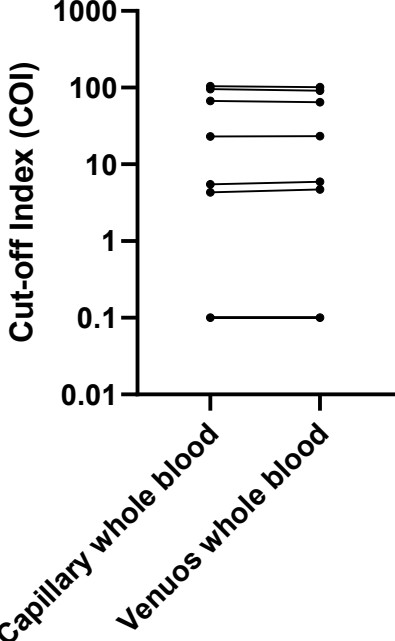

**Figure 1.** Cut-off indices (COIs) from the electrochemiluminescence (ECLIA) SARS-CoV-2 total antibody assay are shown on a semi-logarithmic scale for simultaneously drawn capillary and venous whole blood anticoagulated with EDTA. No significant differences can be observed for the COIs. The line with a COI value of 0.1 was identical for six different participants.

## 3. Methods

### 3.1. Participants

Participants were individuals recruited within the framework of the population-based study in the principality of Liechtenstein. The COVI-GAPP study investigated the role of a sensory bracelet for better and earlier recognition of COVID-19 [15]. Participants with a known history of a COVID-19 positive or negative status were asked to simultaneously provide capillary and venous blood samples. All participants provided written informed consent. The COVI-GAPP study was carried out following the rules of the Declaration of Helsinki of 1975, revised in 2008, and was approved by the Cantonal Ethics Commission of Zürich (KEK Zürich; Basec 2020-00786; 9 April 2020).

### 3.2. Sample Collection

Capillary blood was taken after finger-pricking with safety lancets (Sarstedt, Sevelen, Switzerland) at the lateral side of the middle or ring finger, then collected into Microtainer® MAP microtubes for automated processing (BD, Allschwil, Switzerland). At least 250 µL was sampled from finger-pricking. Venous blood was taken from an antecubital vein right after taking the capillary sample and was collected into a serum Monovette with gel as well as an EDTA-containing Monovette (Sarstedt, Sevelen, Switzerland), following the standard procedures [7].

### 3.3. Laboratory Methods

In the laboratory, hematocrit was determined from the venous EDTA-anticoagulated whole blood by means of a Sysmex XN-1000 (Sysmex, Horgen, Switzerland). After homogenization of the samples on a tube roller, capillary and venous blood samples were analyzed by the electrochemiluminescence method (ECLIA) for the determination of total antibodies directed against the nucleocapsid protein of the SARS-CoV-2 virus (Elecsys® Anti-SARS-CoV-2 assay; Roche Diagnostics, Rotkreuz, Switzerland). The same analysis was also carried out for the serum of the same patients. The results for the EDTA-anticoagulated whole blood were corrected for hematocrit as described in [10]. As provided by the manufacturer, a cut-off index (COI) of 1 and above indicated a positive antibody test for SARS-CoV-2.

### 3.4. Statistical Methods

Data were summarized by either providing the numbers of individuals or the median and interquartile range (IQR). Antibody test results from capillary and venous blood samples were compared by the Wilcoxon test for paired samples. Sample results with $p < 0.05$ were considered to be significant. The data were stored as a Microsoft Excel file (Microsoft, Seattle, WA, USA) as a supplement to this data descriptor (Table S1). Descriptive statistics were calculated with MedCalc version 18.11.3 (MedCalc Software Ltd., Ostend, Belgium). Graphs were drawn with GraphPad Prism version 8.4.3 (GraphPad Software LLC, San Diego, CA, USA).

**Supplementary Materials:** The following are available online at http://www.mdpi.com/2306-5729/5/4/105/s1, Table S1: dataset.

**Author Contributions:** Conceptualization, M.R. (Martin Risch) and L.R.; data curation, M.K.; formal analysis, M.R. (Martin Risch) and M.K.; funding acquisition, L.R.; investigation, M.K.; methodology, M.R. (Martin Risch); project administration, M.A. and C.S.; resources, M.R. (Martin Risch), C.R., D.H., M.R. (Michael Ritzler), N.W., T.L., M.A., C.S. and L.R.; supervision, L.R.; validation, D.H. and N.W.; writing—original draft, M.R. (Martin Risch), M.K., and L.R.; writing—review and editing, M.R. (Martin Risch), M.K., C.R., D.H., M.R. (Michael Ritzler), N.W., T.L., M.A., C.S. and L.R. All authors have read and agreed to the published version of the manuscript.

**Funding:** The COVI-GAPP study was funded by the Liechenstein Government, the Princely House of Liechtenstein, and the Hanela Stiftung, Aarau.

**Conflicts of Interest:** The authors declare no conflict of interest. The funders had no role in the design of the study; in the collection, analyses, or interpretation of data; in the writing of the manuscript, or in the decision to publish the results.

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
