# Peer review of "An Anti-Nucleocapsid Antigen Sars-Cov-2 Total Antibody Assay Finds Comparable Results in Edta-Anticoagulated Whole Blood Obtained from Capillary and Venous Blood Sampling"

_data_

Round 1
Reviewer 1 Report
The existing method of SARS-Co V2 detection involve high throughput laboratory analyzers, and these methods are limited by the requirement of high sample volume. The authors of this manuscript have recently shown that small amounts of capillary blood samples are easy to collect and could prove to be the best sample type when testing regularly, at the population level.
In this manuscript, the authors have pursued the question whether the antibody levels in capillary blood samples are comparable to that of venous blood sampling.
- The authors have collected blood samples from 12 individuals, 6 without and 6 with confirmed COVID-19, and measured the antibody titers (total antibodies directed against the nucleocapsid protein of the SARS-CoV-2 virus) in their blood samples (both capillary and venous) by electro-chemiluminescence (ECLIA) SARS-CoV-2 total antibody assay.
The authors found that the antibody titers between the capillary blood sample and venous blood samples were not significantly different, suggesting that capillary blood samples are indeed an appealing alternative to the existing venous blood samples.
Author Response
The existing method of SARS-Co V2 detection involve high throughput laboratory analyzers, and these methods are limited by the requirement of high sample volume. The authors of this manuscript have recently shown that small amounts of capillary blood samples are easy to collect and could prove to be the best sample type when testing regularly, at the population level.
In this manuscript, the authors have pursued the question whether the antibody levels in capillary blood samples are comparable to that of venous blood sampling.
The authors have collected blood samples from 12 individuals, 6 without and 6 with confirmed COVID-19, and measured the antibody titers (total antibodies directed against the nucleocapsid protein of the SARS-CoV-2 virus) in their blood samples (both capillary and venous) by electro-chemiluminescence (ECLIA) SARS-CoV-2 total antibody assay.
The authors found that the antibody titers between the capillary blood sample and venous blood samples were not significantly different, suggesting that capillary blood samples are indeed an appealing alternative to the existing venous blood samples.
Thank you for appreciating our work.
Reviewer 2 Report
The present study focuses on the comparability of anti-SARS-CoV-2 antibody assay results from capillary and venous blood sampling. Although this topic is interesting, the number of samples included in the study is very limited and comparison to other serological SARS-CoV-2 assays including different target structures is missing.
Author Response
The present study focuses on the comparability of anti-SARS-CoV-2 antibody assay results from capillary and venous blood sampling.
Thank you for you review.
Although this topic is interesting, the number of samples included in the study is very limited and comparison to other serological SARS-CoV-2 assays including different target structures is missing.
Due to the low volume of capillary blood sampling we were limited relating to other test formats. We therefore studied the assay, which is used as a routine test in our lab.
Regarding the number of samples, we have noted a limitation in the data description: “ As the number of samples is limited, minor differences between capillary and venous blood might go unnoticed due to statistical type II error.”
Reviewer 3 Report
This paper describes/documents the antiviral antibody results with SARS-CoV-2 (the cause of COVID-19) as obtained with the "regular" sample type (serum of venous blood) cf. finger-prick-derived whole blood. The results in the two blood-sample types are virtually identical, unsurprisingly.
The execution and presentation of this study are excellent. The practical importance and journalistic interest of the subject (in light of its topicality) are higher than its academic/intellectual novelty. I.e., more than 10 similar/corresponding studies have been done over decades with other human viruses, with similar results. Two are appropriately cited.
One very recent publication (attached) does call for comparative recognition and citation: McDade et al, High seroprevalence for SARS-CoV-2 among household members of essential workers detected using a dried blood spot assay. PLoS ONE August, 2020 (https://doi.org/10.1371/journal.pone.0237833).

Author Response
This paper describes/documents the antiviral antibody results with SARS-CoV-2 (the cause of COVID-19) as obtained with the "regular" sample type (serum of venous blood) cf. finger-prick-derived whole blood. The results in the two blood-sample types are virtually identical, unsurprisingly.
The execution and presentation of this study are excellent.
Thank you for appreciating our work.
The practical importance and journalistic interest of the subject (in light of its topicality) are higher than its academic/intellectual novelty. I.e., more than 10 similar/corresponding studies have been done over decades with other human viruses, with similar results. Two are appropriately cited. One very recent publication (attached) does call for comparative recognition and citation: McDade et al, High seroprevalence for SARS-CoV-2 among household members of essential workers detected using a dried blood spot assay. PLoS ONE August, 2020 (https://doi.org/10.1371/journal.pone.0237833).
Thank you for this suggestion. We have added this reference to the reference list and added a sentence as follows: “Another study found that results for an ELISA assay detecting IgG antibodies directed against the receptor binding domain of the S1 protein are comparable between serum from venous blood drawing and dried blood spots obtained from capillary blood sampling [9].”